# The global Higgs picture at 27 TeV

**Anke Biekötter[1], Dorival Gonçalves[2], Tilman Plehn[1],**
**Michihisa Takeuchi[3] and Dirk Zerwas[4]**

**1** Institut für Theoretische Physik, Universität Heidelberg, Germany
**2** PITT PACC, Department of Physics and Astronomy, University of Pittsburgh, USA
**3** Kavli IPMU (WPI), UTIAS, University of Tokyo, Kashiwa, Japan
**4** LAL, IN2P3/CNRS/ Orsay, France

⋆ biekoetter@thphys.uni-heidelberg.de

## Abstract

**We estimate the reach of global Higgs analyses at a 27 TeV hadron collider in terms of Higgs couplings and in terms of a gauge-invariant effective Lagrangian, including invisible Higgs decays and the Higgs self-coupling. The new collider will indirectly probe new physics in the TeV range and allow for a meaningful test of the Higgs self-coupling also embedded in a global analysis.**

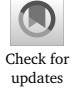

**Content**

## 1 Introduction

In record time Higgs physics has moved from a spectacular discovery of a new particle to a systematic and comprehensive study of its properties [1]. The general development of hadron collider physics into precision physics has been fueled by the understanding of the LHC detectors, the control of perturbative QCD, the ability to precisely simulate even complex LHC

processes, and the theoretical understanding of a perturbative electroweak Lagrangian as an interpretation framework. This naturally leads to the question what kind of precision we can reach with a 27 TeV hadron collider with an attobarn-level integrated luminosity.

If we assume that the observed scalar is really Higgs-like, specifically that it is responsible for electroweak symmetry breaking and hence forms a doublet with the weak Goldstone modes, we can interpret the LHC results in terms of an effective Lagrangian with linearly realized electroweak symmetry breaking [2–11]. This defines the framework of some of the most interesting global Higgs analyses based on Run I [12–15] and even Run II [16] data. Because this effective Lagrangian firmly links the Higgs and electroweak sectors, the global analysis has to incorporate anomalous triple gauge boson measurements from LEP [17, 18] and the LHC [12, 18–23]. At the 13 TeV LHC the Higgs self-coupling can be neglected at the typical precision of a global Higgs analysis [24]. In contrast, a 27 TeV hadron collider is expected to contribute a meaningful measurement of Higgs pair production [25–28], for example testing a possible first-order electroweak phase transition as an ingredient to baryogenesis [29, 30]. Finally, we include invisible Higgs decays in terms of an invisible branching ratio following Ref. [31].

For our brief study we start from the established Run I limits of the LHC [12, 32] and extrapolate them to an upgraded LHC setup*. This provides a reliable benchmark for such an energy upgrade with a large integrated luminosity. First, we discuss the 27 TeV projections in terms of Higgs coupling modifiers, motivated by an effective Lagrangian with a non-linear realization of electroweak symmetry breaking, in Sec. 2. In Sec. 3 we then use a linear realization to combine the global Higgs analysis with di-boson data. In Sec. 4 we discuss the benefits from including a differential measurement of Higgs pair production in detail.

Eventually, this analysis should be combined with anomalous gauge couplings to fermions, as well as electroweak precision data [16, 22, 23, 34–36]. We leave such a more detailed analysis also including a larger set of kinematic distributions [37] and an appropriate validation on ATLAS and CMS results for the future. For now, this extrapolation based on our established and validated 8 TeV analysis will provide a reliable first estimate. Such a conservative treatment is in order, because dependent on the interpretation framework the global Higgs analyses at a 27 TeV collider will rapidly enter systematics-limited and theory-limited territory.

Table 1: Relative theory uncertainties for the different production and decay channels contributing to the global analysis. The numbers correspond to those quoted in Ref. [38].

| production | [%] | decay | [%] |
|---:|---:|---:|---:|
| GF | 10.2 | $WW$ | 2.63 |
| $qqH$ | 3.0 | $ZZ$ | 2.63 |
| $WH$ | 3.2 | $\gamma\gamma$ | 3.31 |
| $ZH$ | 5.7 | $b\bar{b}$ | 2.17 |
| $t\bar{t}H$ | 12.8 | $Z\gamma$ | 7.33 |
| $HH$ | 18. | $\tau\tau$ | 2.78 |

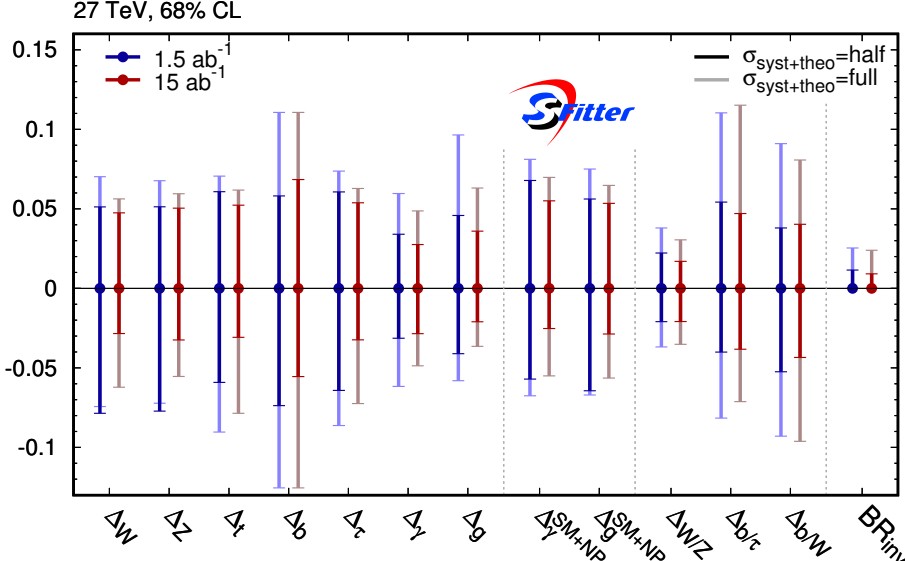

Figure 1: Result from the global Higgs analysis in terms of coupling modifiers or non-linearly realized electroweak symmetry breaking. All limits are shown as profiled over all other couplings.

## 2 Global Higgs analysis

Historically [33, 39, 40], new physics effects on the SM-like Higgs couplings have been parameterized as coupling modifiers

$$g_x = g_x^{\text{SM}} (1 + \Delta_x)$$
$$g_{g,\gamma} = g_{g,\gamma}^{\text{SM}} (1 + \Delta_{g,\gamma}^{\text{SM}} + \Delta_{g,\gamma}) \equiv g_{g,\gamma}^{\text{SM}} (1 + \Delta_{g,\gamma}^{\text{SM+NP}}),$$
(1)

where the $\Delta_x$ can be directly translated into the experimentally used $\kappa$ notation

$$\kappa_x = (1 + \Delta_x)$$
(2)

at least modulo the treatment of the tree-level couplings contributing to the loop-induced operators. The corresponding global Higgs analysis is the main reason why we can now claim that the observed Higgs boson closely follows the Standard Model predictions. In terms of a Lagrangian we can write this hypothesis as [12]

$$\mathcal{L} = \mathcal{L}_{\text{SM}} + \Delta_W \, g m_W h \, W^\mu W_\mu + \Delta_Z \, \frac{g}{2c_W} m_Z h \, Z^\mu Z_\mu - \sum_{\tau,b,t} \Delta_f \, \frac{m_f}{v} h \left( \bar{f}_R f_L + \text{h.c.} \right)$$

$$+ \Delta_g F_G \, \frac{h}{v} \, G_{\mu\nu} G^{\mu\nu} + \Delta_\gamma F_A \, \frac{h}{v} \, A_{\mu\nu} A^{\mu\nu} + \text{invisible decays} .$$
(3)

This Lagrangian shifts all numerical values of the SM-like Higgs couplings and breaks electroweak gauge invariance. The modified dimension-4 coupling terms obviously affect all loop-induced Higgs couplings. In addition, the Lagrangian includes new higher-dimensional operators coupling the Higgs to photons and gluons. They arise from potential new particles in the loop and are normalized to their Standard Model values $F_G$ and $F_A$. In the limit of heavy top masses these normalization constants read $F_G^{(\infty)} \to \alpha_s/(12\pi)$. We refrain from including

---

*Many aspects of our 27 TeV study are described in detail in these 8 TeV legacy papers [12, 32], including a validation of the 8 TeV results. The SFITTER error treatment is discussed in Ref. [33].

a more complete set of operators, to be consistent with existing analyses. The form of Eq.(3) can be trivially mapped onto an effective Lagrangian with a non-linear representation of the Higgs and Goldstone fields, resulting in a broken $SU(2)$ doublet structure [17, 41–43].

Including invisible Higgs decays in terms of an effective Lagrangian would force us to either define a new particle with unknown quantum numbers, or to re-scale the decay $H \rightarrow 4\nu$ to ridiculous branching ratios. Instead, we include invisible Higgs decays in terms of the corresponding partial width or, equivalently, the invisible branching ratio. The total Higgs width is consistently constructed out of all observed partial width, with an assumed scaling of the second and third generation of Yukawa couplings, as described in Ref. [12].

In principle it would be possible to include the Higgs self-coupling in the global, non-linear Higgs analysis. However, we know that di-Higgs measurements will not improve any of the parameters given in Eq.(3) and that the Higgs self-coupling does not affect single-Higgs production in a relevant way. Because the extration of the Higgs self-coupling crucially depends on kinematic distributions [24, 25] we postpone this aspect to Sec. 3, where we also include a full set of di-boson and single Higgs distributions.

The global Higgs analysis in terms of Eq.(3) only describes total cross sections in the Higgs sector. We therefore just re-scale the number of signal and background events in the 8 TeV analysis [12] to 27 TeV, assuming two experiments. This affects all statistical uncertainties, as well as the systematics, which we assume to be related to measurements in control regions. For all measurements we assume the SM predictions, which means that our best-fit points will always be the SM values. For the invisible Higgs searches we use an in-house extrapolation of the WBF analysis from Ref. [31] to 27 TeV. The current theory uncertainties of all measurements are listed in Tab. 1, including uncertainties on the parton distributions. For these, we simply assume that dedicated fits will determine the PDFs at a 27 TeV collider will full luminosity at the same level as they are determined for the LHC now. To illustrate the point that precision predictions and PDF extraction are crucial we will show results with the current theory uncertainties as well as an assumed improvement of theory and systematics by a factor two.

In Fig. 1 we show the expected precision of the SM-like Higgs coupling measurements for a 27 TeV LHC upgrade. Asymmetric uncertainty bands arise because of correlations, but also reflect numerical uncertainties. Different colors correspond to assumed integrated luminosities of 1.5 ab$^{-1}$ and 15 ab$^{-1}$. For all coupling deviations, with the exception of $\Delta_b$, we observe an improvement with increased luminosity. However, this improvement is much smaller than the rough factor three which one could expect from a scaling of the limits with the square-root of the luminosity, indicating that the limits are systematics and theory limited. Ratios of couplings, like $\Delta_{W/Z}$, see little improvement from an increased luminosity. To confirm the domination by systematics and theory uncertainties, we also compare today's theory and systematic uncertainties with an improvement to half the current uncertainties indicated by full and shaded bands. For example for $\Delta_b$ and $\Delta_g$, as well as for the coupling ratios we indeed see a significant improvement. Altogether, the typical precision in measuring Higgs couplings can reach 3% to 5% at a 27 TeV hadron collider with a realistic improvement of the systematic and theory uncertainties. The ratio of the $W$ and $Z$ couplings to the Higgs, for example, will benefit from correlated uncertainties and will therefore be measured a factor two more precisely than the individual couplings. Invisible Higgs decays will be constrained at the branching ratio level of 1% to 2%. Compared to the high-luminosity LHC predictions in Ref. [44] the 27 TeV with its projected final luminosity will double the precision on many of the Higgs coupling modifications.

# 3 Higgs-gauge analysis

An effective Lagrangian is defined by its particle content and its symmetries, with the dimensionality of the individual operators, or inverse powers of a large matching scale as the expansion parameter. Truncated to dimension six it has the form [6, 9]

$$\mathcal{L} = \sum_x \frac{f_x}{\Lambda^2} \mathcal{O}_x \,, \tag{4}$$

where $\Lambda \gg v$ is the scale of the assumed UV-complete model. It can be extended to the full Standard Model particle content, defining the Standard Model Effective Field Theory [11]. The minimum independent set of dimension-6 operators with the SM particle content, compatible with the SM gauge symmetries, and compatible with baryon number conservation contains 59 operators [9]. We impose $C$ and $P$ invariance [45] on the operator set of Ref. [6], use the equations of motion including all necessary fermionic operators to avoid blind directions from electroweak precision data, and neglect all operators that will not be constrained by LHC Higgs measurements. For the Higgs couplings to fermions we assume a Yukawa coupling structure, noting that current LHC analyses are unlikely to test the Lorentz structure of a possible deviation from the Standard Model. This gives us the Lagrangian

$$\begin{aligned}
\mathcal{L}_{\text{eff}} = &-\frac{\alpha_s}{8\pi}\frac{f_{GG}}{\Lambda^2}\mathcal{O}_{GG} + \frac{f_{BB}}{\Lambda^2}\mathcal{O}_{BB} + \frac{f_{WW}}{\Lambda^2}\mathcal{O}_{WW} + \frac{f_B}{\Lambda^2}\mathcal{O}_B + \frac{f_W}{\Lambda^2}\mathcal{O}_W + \frac{f_{WWW}}{\Lambda^2}\mathcal{O}_{WWW} \\
&+ \frac{f_{\phi 2}}{\Lambda^2}\mathcal{O}_{\phi 2} + \frac{f_{\phi 3}}{\Lambda^2}\mathcal{O}_{\phi 3} + \frac{f_\tau m_\tau}{v\Lambda^2}\mathcal{O}_{e\phi,33} + \frac{f_b m_b}{v\Lambda^2}\mathcal{O}_{d\phi,33} + \frac{f_t m_t}{v\Lambda^2}\mathcal{O}_{u\phi,33} \\
&+ \text{invisible decays}\,,
\end{aligned} \tag{5}$$

with the operators defined as in Ref. [32],

$$\begin{aligned}
\mathcal{O}_{GG} &= \phi^\dagger\phi\, G^a_{\mu\nu}G^{a\mu\nu} & \mathcal{O}_{BB} &= \phi^\dagger\hat{B}_{\mu\nu}\hat{B}^{\mu\nu}\phi & \mathcal{O}_{WW} &= \phi^\dagger\hat{W}_{\mu\nu}\hat{W}^{\mu\nu}\phi \\
\mathcal{O}_B &= (D_\mu\phi)^\dagger\hat{B}^{\mu\nu}(D_\nu\phi) & \mathcal{O}_W &= (D_\mu\phi)^\dagger\hat{W}^{\mu\nu}(D_\nu\phi) & \mathcal{O}_{WWW} &= \text{Tr}\left(\hat{W}_{\mu\nu}\hat{W}^{\nu\rho}\hat{W}^\mu_\rho\right) \\
\mathcal{O}_{\phi 2} &= \frac{1}{2}\partial^\mu(\phi^\dagger\phi)\partial_\mu(\phi^\dagger\phi) & \mathcal{O}_{\phi 3} &= -(\phi^\dagger\phi)^3/3 \\
\mathcal{O}_{e\phi,33} &= \phi^\dagger\phi\, \bar{L}_3\phi e_{R,3} & \mathcal{O}_{d\phi,33} &= \phi^\dagger\phi\, \bar{Q}_3\phi d_{R,3} & \mathcal{O}_{u\phi,33} &= \phi^\dagger\phi\, \bar{Q}_3\tilde{\phi}u_{R,3}\,.
\end{aligned} \tag{6}$$

The pure gauge operator $\mathcal{O}_{WWW}$ is needed to fully describe anomalous triple gauge couplings in a gauge-invariant framework. Any LHC analysis also needs to include an anomalous triple gluon interaction, but this operator is constrained by multi-jet production at 13 TeV much more strongly than any Higgs analysis will achieve [46], and we assume that this pattern will be the same at a 27 TeV collider. Recent studies [23, 36, 47] have shown the relevance of fermionic operators through their induced $f\bar{f}V$, $f\bar{f}VV$ and $f\bar{f}VH$ couplings, which require a combination of the Higgs–gauge analysis with electoweak precision data. Such an analysis is beyond the scope of this projection, but eventually the fermionic operators should be part of a global SMEFT fit.

Because the effective Lagrangian of Eq.(5) includes new Lorentz structures, especially valuable information comes from kinematic distributions probing interactions with a large momentum flow. We therefore include four single Higgs and four di-boson distributions in our analysis [32]. For the 8 TeV analysis they are validated with existing data [32, 48, 49]. We use MAD-GRAPH5-2.3.2.2 [50], PYTHIA6-2.4.5 [51], and DELPHES3.1.2 [52] for two ATLAS/CMS-like experiments. For distributions, where different cuts define different phase space regions, we only use the high-momentum regime. Finally, we include Higgs pair production $pp \to HH \to b\bar{b}\gamma\gamma$ including kinematic information in terms of $m_{HH}$, as pioneered in Ref. [53], now accounting

for two different jet multiplicities [25]. All distributions entering our analysis are summarized in Tab. 2, the details of the Higgs pair production process will be discussed in Sec. 4.

Comparing the reach of the main kinematic distributions in Tab. 2 we see that the $VV$ channels probe a much larger momentum flow than the $VH$ channels. This can be traced to the larger signal rates, namely $\sigma_{WZ} = 61.1$ pb vs $\sigma_{WH} = 2.8$ pb at 27 TeV and to leading order [50], combined with higher tails through the momentum-dependent $WWZ$ coupling. For example comparing $WZ$ production with $WH$ production, the reach in $p_T^V$ is defined by the highest bins with a sizeable number of signal events. Specifically, we ignore phase space regions with fewer than three signal events for an integrated luminosity of 15 ab$^{-1}$.

In Fig. 2 we show the results of the global Higgs analysis in terms of dimension-6 SMEFT operators, including the quadratic terms of the EFT expansion. The right axis indicates the new-physics scale $\Lambda$ assuming reasonably strongly interacting new physics $f_x = 1$. The 27 TeV analysis is typically sensitive to new-physics scales well above 1 TeV. This number should be compared to the range of distributions given in Tab. 2, indicating that our Higgs analysis does not have any serious EFT validity issues provided we do not see a pole in the diboson channels. The asymmetric error bands for some of the Wilson coefficients will be discussed in the next section.

The balance of statistical, systematic, and theory uncertainties in the SMEFT analysis is significantly different from the non-linear coupling modifiers shown in Fig. 1. Effective operators benefit from an increased statistics, because larger luminosity extends the reach of kinematic distributions, which in their tails are always statistically limited. In contrast, the Yukawa couplings $f_{b,\tau}$, which do not change the Lorentz structure, are mostly limited by the assumed systematic and theory uncertainties. Consequently, the reach for operators which modify the Lorentz structure of some Higgs interaction exceeds the reach for the Yukawa-like operators or the reach for the operator $\mathcal{O}_{\phi 2}$, which introduces a wave function renormalization for the Higgs field and only changes the kinematics of Higgs pair production. For the former the kinematic distributions drive the limits towards $\Lambda/\sqrt{f} \approx 3$ TeV and beyond, for high luminosity and improved systematics and theory uncertainties. This aspect is where we also expect significant improvements from a dedicated 27 TeV study developing analysis ideas not realized at 8 TeV.

The asymmetric limits on $f_B$ give us some insight into the structure of the effective theory. This operator is largely constrained through $VH$ production at high momentum transfer, specifically the $p_T^V$ distributions from Tab. 2. In its highest available bins we probe sizeable

Table 2: Distributions included in the analysis. The number of bins includes an overflow bin for all channels.

| channel | observable | # bins | range [GeV] |
|---|---|---|---|
| $WW \to (\ell\nu)(\ell\nu)$ | $m_{\ell\ell'}$ | 10 | $0 - 4500$ |
| $WW \to (\ell\nu)(\ell\nu)$ | $p_T^{\ell_1}$ | 8 | $0 - 1750$ |
| $WZ \to (\ell\nu)(\ell\ell)$ | $m_T^{WZ}$ | 11 | $0 - 5000$ |
| $WZ \to (\ell\nu)(\ell\ell)$ | $p_T^{\ell\ell}$ $(p_T^Z)$ | 9 | $0 - 2400$ |
| WBF, $H \to \gamma\gamma$ | $p_T^{\ell_1}$ | 9 | $0 - 2400$ |
| $VH \to (0\ell)(b\bar{b})$ | $p_T^V$ | 7 | $150 - 750$ |
| $VH \to (1\ell)(b\bar{b})$ | $p_T^V$ | 7 | $150 - 750$ |
| $VH \to (2\ell)(b\bar{b})$ | $p_T^V$ | 7 | $150 - 750$ |
| $HH \to (b\bar{b})(\gamma\gamma)$, 2$j$ | $m_{HH}$ | 9 | $200 - 1000$ |
| $HH \to (b\bar{b})(\gamma\gamma)$, 3$j$ | $m_{HH}$ | 9 | $200 - 1000$ |

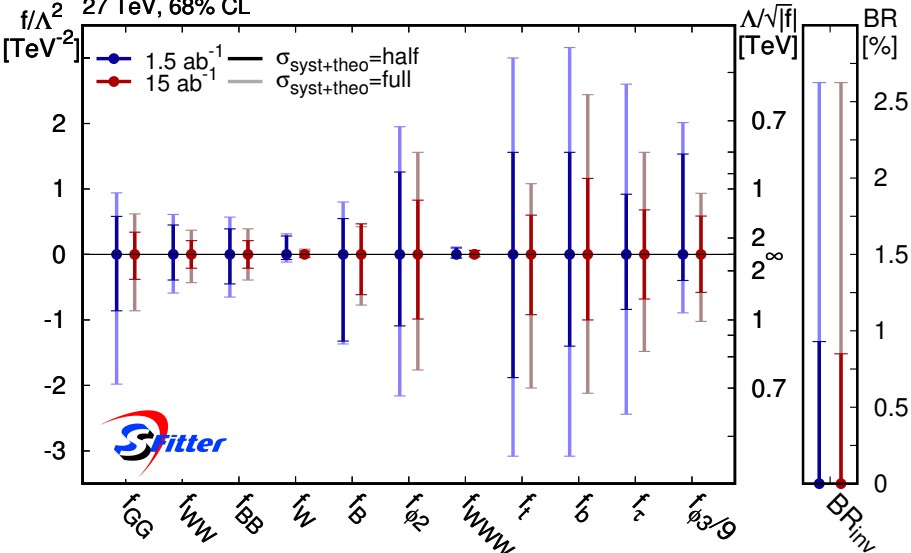

Figure 2: Result from the global Higgs analysis in terms of dimension-6 operators. All limits are shown as profiled over all other Wilson coefficients.

ratios $p_T/\Lambda$, but with sizeable statistical uncertainties. If we include the dimension-6 squared terms a second solution predicting the same event count within the statistical uncertainties appears for $f_B/\Lambda > 0$. For this second solution the squared term compensates a small destructive interference with the SM contribution. Because the precise position of this secondary solution differs for different values of $p_{T,V}$, it induces a slightly asymmetric measurement of $f_B/\Lambda$. Note that a visible dimension-6-squared term in a specific observable does by no means signal the breakdown of the effective Lagrangian [54]. The validity of an effective field theory representing classes of underlying UV-complete models can only be judged once we identify on-shell contributions of the new particles [55]. Second, truncating the expansion of our observables after the linear term in $f/\Lambda^2$ would lead to a symmetric and more narrow likelihood distribution and underestimate of the errors. In general, we do not include uncertainties on the EFT framework in our global analysis, as we consider them to be uncertainties on the matching and interpretation of our results in terms of a UV complete model [56, 57].

## 4 Higgs self-interaction

An enhanced Higgs self-coupling as a simple modification of the SM Higgs potential is especially interesting for example in relation to vacuum stability and baryogenesis [29, 30]. Including it in our global Higgs analysis is a significant improvement as compared to the Run I legacy analysis [32]. It is made possible by the fact that a 27 TeV collider with a large integrated luminosity will allow for a dedicated measurement of the Higgs self-coupling. The self-coupling with its unique relation to the Higgs potential is not yet included in most global analyses of SM-like Higgs couplings because of the modest reach of the LHC. However, for a 27 TeV collider with an integrated luminosity of 15 ab$^{-1}$ we quote the expected reach [25]

$$\frac{\lambda_{3H}}{\lambda_{3H}^{(SM)}} = \begin{cases} 1 \pm 15\% & 68\% \text{ C.L.} \\ 1 \pm 30\% & 95\% \text{ C.L.} \end{cases} \tag{7}$$

We can translate this range into the conventions of Eq.(5) if we assume that the underlying new physics does not generate any other dimension-6 operator. In that case we assume

$$V = \mu^2 \frac{(v+H)^2}{2} + \lambda \frac{(v+H)^4}{4} + \frac{f_{\phi 3}}{3\Lambda^2} \frac{(v+H)^6}{8} , \tag{8}$$

and find for the reach of the dedicated self-coupling analysis [58]

$$\lambda_{3H} = \lambda_{3H}^{(\text{SM})} \left( 1 + \frac{2v^2}{3m_H^2} \frac{f_{\phi 3} v^2}{\Lambda^2} \right) \quad \text{and} \quad \left| \frac{\Lambda}{\sqrt{f_{\phi 3}}} \right| \gtrsim \begin{cases} 1 \text{ TeV} & 68\% \text{ C.L.} \\ 700 \text{ GeV} & 95\% \text{ C.L.} \end{cases} \tag{9}$$

While we are free to define a modified Higgs potential as our physics hypothesis [30], this setup is in direct violation of the effective Lagrangian approach. Here all operators consistent with the symmetry assumptions have to be included. Consequently, $\mathcal{O}_{\phi 2}$ also affects the Higgs pair production process with a momentum-dependent self-coupling [58,59].

The full kinematic information from Higgs pair production encoded in the $m_{HH}$ distribution allows us to separate the effects of $\mathcal{O}_{\phi 2}$ and $\mathcal{O}_{\phi 3}$ [24,25]. In addition, we can use single Higgs production to constrain $\mathcal{O}_{GG}$ and $\mathcal{O}_{u\phi,33}$, as seen in Sec. 3. All corresponding error bands have to be propagated into Higgs pair production, and given the size of the uncertainties we can safely assume that in the presence of $\mathcal{O}_{\phi 3}$ Higgs pair production will hardly help with any of the operators already constrained by single Higgs production. Following Refs. [60,61] and especially Ref. [62] we also neglect the loop effects of $\mathcal{O}_{\phi 3}$ on single Higgs production, because they will hardly affect a global Higgs analysis. Finally, looking at different uncertainties it is also obvious that a study of the $m_{HH}$ distribution will be statistically limited even at the 27 TeV collider, in contrast to the typical total rate measurements discussed before.

In Fig. 3 we illustrate the correlation of $\mathcal{O}_{\phi 2}$ and $\mathcal{O}_{\phi 3}$ through the full expression for a modified self-coupling [58]. We also know that positive and negative deviations of the Higgs self-coupling affect different phase space regions [24]: while a reduced value of the self-coupling can be tested around the threshold $m_{HH} \approx 2m_H$, an increase in the self-coupling requires us to look for large values of $m_{HH}$ and to deal with large effects from $\mathcal{O}_{\phi 2}$. All of this leads to an asymmetric uncertainty band on $\mathcal{O}_{\phi 3}$ especially once we allow for an agreement of the SM-like measurements with the SM predictions at two sigma and integrate the asymmetric tails

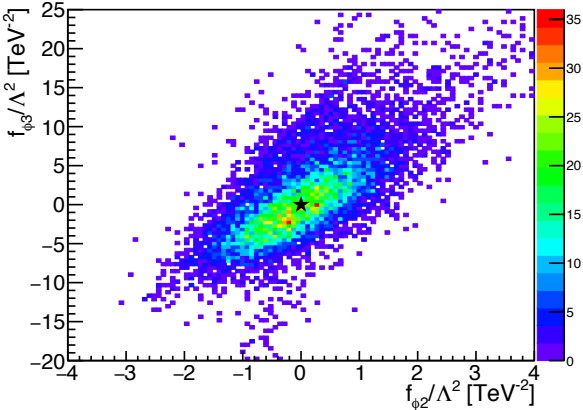

Figure 3: Correlations between the leading operators describing Higgs pair production.

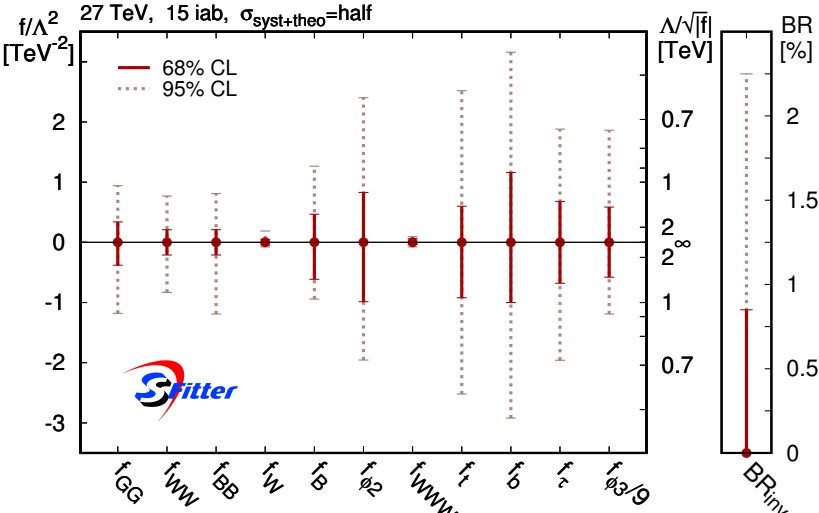

Figure 4: Result from the global Higgs analysis in terms of dimension-6 operators, complementing the high-luminosity and improved-error scenario of Fig. 2 with results for 95% C.L. or two standard deviations.

further. From Fig. 4 we read off the limits

$$\frac{\Lambda}{\sqrt{|f_{\phi 3}|}} > 430 \text{ GeV} \qquad\qquad\qquad\qquad\qquad\qquad 68\% \text{ C.L.}$$

$$\frac{\Lambda}{\sqrt{|f_{\phi 3}|}} > 245 \text{ GeV} \quad (f_{\phi 3} > 0) \quad \text{and} \quad \frac{\Lambda}{\sqrt{|f_{\phi 3}|}} > 300 \text{ GeV} \quad (f_{\phi 3} < 0) \qquad 95\% \text{ C.L.} \quad (10)$$

These limits are diluted from the one-parameter analysis quoted in Eq.(9), largely because of the combination with $\mathcal{O}_{\phi 2}$. As a matter of fact, we can directly compare the effects from $\mathcal{O}_{\phi 2}$ and $\mathcal{O}_{\phi 3}$ for similar values of $f/\Lambda^2$ as a function of the momentum flowing through the triple-Higgs vertex or $m_{HH}$. In that case we find that the momentum dependence in $\mathcal{O}_{\phi 2}$ matches the effects from $\mathcal{O}_{\phi 3}$ for $m_{HH} \gtrsim 1$ TeV, with either relative sign. This additional source of a modified self-coupling characterized by the interplay between $\mathcal{O}_{\phi 2}$ and $\mathcal{O}_{\phi 3}$ is not accounted for in the usual Higgs pair analyses.

## 5  Outlook

Following the established Run I legacy results [12, 32] we estimate the reach of a 27 TeV hadron collider in a global analysis of the Higgs-gauge sector. We include not only invisible Higgs decays, but also Higgs pair production, sensitive to the Higgs self-coupling.

First, we interpret the extrapolated measurement in terms of modified SM-like Higgs couplings, motivated by an effective theory with non-linearly realized electroweak symmetry breaking. We find that a 27 TeV hadron collider will be sensitive to 3 ... 5% deviations from the SM coupling values. Systematics and theory uncertainties rapidly limit the reach for modified Higgs couplings beyond attobarn-level integrated luminosities.

Using a gauge-invariant effective theory in terms of the Higgs doublet allows us to include di-boson rates and kinematic distributions in the global Higgs analysis. Invisible branching ratios can be extracted from a global Higgs analysis to better than one per-cent. For the gauge-invariant interpretation framework we also include a modified Higgs potential at dimension six. The additional Wilson coefficient can be constrained by a kinematic analysis of Higgs pair

production [25] if we control the correlation with the operator $\mathcal{O}_{\phi 2}$.

We find a TeV-scale reach for new physics given order-one Wilson coefficients for most dimension-6 operators. Those operators which change the Lorentz structure of Higgs couplings can be strongly constrained beyond the 3 TeV level. Comparing those numbers with the current reach of the LHC Run II [16, 35, 36], we find that the reach of a 27 TeV hadron collider could increase the bounds on the new physics scale by more than 50%. The global analysis obviously reduces the reach for the Higgs self-coupling modification compared to one-parameter analysis, but still indicates that a 27 TeV hadron collider will for the first time deliver a meaningful measurement of this fundamental physics parameter.

## Acknowledgments

First and foremost we are grateful to Michael Rauch, not only because of his significant contribution in the early phase of this project, but also because of his long-term leading role in all SFitter analyses. If our field decides to let technically cutting-edge people like him go, we should not be surprised that large parts of BSM theory are losing contact to LHC data and are becoming increasingly irrelevant to LHC physics as a whole.

We would like to thank Jennie Thompson for providing the numbers for the invisible Higgs analysis. AB is funded by the DFG through the Graduiertenkolleg Particle physics beyond the Standard Model (GRK 1940) and the IMPRS-PTFS. The authors acknowledge support by the state of Baden-Württemberg through bwHPC and the German Research Foundation (DFG) through grant no INST 39/963-1 FUGG (bwForCluster NEMO). MT is supported in part by the JSPS Grant-in-Aid for Scientific Research Numbers 16H03991, 16H02176, 17H05399, 18K03611, and the World Premier International Research Center Initiative, MEXT, Japan.

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
