# Peer review of "The Global Higgs Picture at 27 TeV"

_SciPost Physics, doi:SciPost Phys. 6, 024 (2019)_

## Round 1 · Referee Report · Anonymous (Referee 1) · 2018-12-20

Strengths

1 - the paper addresses a very interesting question, that had not been explored in detail before. This study is therefore very welcome.
2 - The numerical analysis is rich: several classes of measurements are considered, including kinematic distributions and di-Higgs production
3 - The analysis is developed within Sfitter, that ensures a very good quality fit
4 - The paper is very clear in every part.

Weaknesses

1 - The work presented is valid and more than sufficient for a first estimate, but there is room for improvement and refinement of the study, e.g. including more parameters / operators.

Report

The authors investigate the impact of Higgs and electroweak measurements at a 27 GeV hadron collider on global Higgs analyses, realized in the context of the SM EFT or in terms of Higgs anomalous couplings. Although the main goal of such a machine (that has been proposed as a future High Energy phase of the LHC) would be to search for heavy resonances that could not be detected at the LHC, a scenario without new discoveries can not be excluded a priori. In this case, incorporating new high-energy data into LHC EFT analyses would be extremely important.
The analysis presented here brings then valuable information to the physics case of the HE-LHC.

The manuscript is clear and well written. I recommend it for publication once the issues below are addressed.

Requested changes

Questions / requests in order of appearance in the paper:

1 - after eq (2). The authors state that "a more complete set of loop-induced operators [...] would compete with finite tree-level values and are phenomenologically less relevant".
I believe they are referring to the structures $hZ_{\mu\nu}^2$, $hW_{\mu\nu}^2$.

It's not clear to me why these terms are phenomenologically less relevant than those in eq (2): in fact they are directly related by gauge invariance to Delta_gamma, and, they can pick up a kinematic enhancement due to extra momentum dependence compared to the SM coupling structure. This is in fact shown to be the case for the OWW, OBB operators, that are included in the EFT fit. What is the difference between the $hZ_{\mu\nu}^2$, $hW_{\mu\nu}^2$ terms and the anomalous couplings induced by OWW and OBB, that motivates a different treatment?

Also, as a minor comment, the characterization as tree-induced and loop-induced structures is known not to be well-defined in general and possibly violated, in particular, in the presence of strong interacting new physics. I would just advise the authors to omit this distinction.

2 - in the same paragraph, the wording "resulting in a broken SU(2) gauge invariance" is incorrect and (if I'm interpreting correctly) should be rephrased stating that the SU(2) doublet structure of the Higgs field is explicitly broken in the non-linear Lagrangian. Not gauge invariance itself.

3 - two paragraphs below the authors state that "the non-linear Lagrangian does not describe such a change in distributions". However, the singlet nature of the Higgs in the non-linear Lagrangian allows for many independent Lorentz structures in the H self-coupling, in fact more than in the linear Lagrangian. What are the authors referring to here?

4 - eq (4). It's not clear while none of the fermionic operators modifying $Z\bar ff$/$W\bar ff$ couplings were retained, as the explanation given just above eq (4) does not seem to be a sufficient justification. Applying the EOM allows to remove only a subset of them in favor of bosonic operators, and they can definitely be constrained in LHC Higgs measurements (they enter e.g. $h \rightarrow 4\ell$).

5 - I am concerned about the choice of using "only the high-momentum regime" of kinematic distributions. The reason is two-fold: on one hand this regime can be plagued by EFT validity issues. The authors make a check a posteriori and conclude that their bounds (for f=1) are indeed consistent. However, as far as I can read from table 2 and fig 2, the reach in lambda never exceeds 2 TeV, while distributions up to 4 TeV are considered (fBB, fWW). But then using the bins between 2-4 TeV is naively inconsistent (one is saying that the signal at 4 TeV is modeled with a lagrangian valid up to 2 TeV). How were these bins treated?
The second reason is related to the estimate of theoretical uncertainties in this region, that I address in the following point.

6 - The analysis results rely on some assumptions on the theoretical uncertainties for Higgs measurements, that are always taken to be $\lesssim 10\%$. However, there are at least two sources of uncertainties that could have a larger impact, especially on the constraints dominated by measurements in high-momentum regions, and therefore deserve some discussion:
a) depending on the initial parton state, PDF uncertainties can be very large (even ~100%) at large $x$. Did the authors check how they would change at the new collider? Is it realistic to expect that they will remain equal or smaller than those at the LHC, even in the very high-momentum regions considered here?
b) the EFT formalism comes in principle with its own theoretical uncertainty, that accounts for radiative corrections as well as for the effect of truncating the expansion (in this case $d\geq 8$ operators and double $d=6$ insertions in the amplitude). By dimensional analysis, it should be at least of order $(E/\Lambda)^4$ or higher. For $\Lambda\sim 10$ TeV this can easily be ~10% at E=4 TeV. How would this impact the constraints and/or the EFT validity of the results?

7 - finally, a minor point: it would be nice to comment briefly in the conclusions on how the estimated reach for a 27 TeV collider compares to the projected reach for direct searches, as well as to the current bounds from LHC (how large an improvement will this bring?).

  • validity: high
  • significance: high
  • originality: top
  • clarity: high
  • formatting: excellent
  • grammar: perfect

Author:  Anke Biekoetter  on 2019-01-17  [id 405]

(in reply to Report 1 on 2018-12-20)

Dear Referee,

Thank you very much for your report and your very useful comments.

We have addressed your concern in version 2 of our paper in the following way:

1 See (1) in reply to report 2

2 We rephrased the sentence.

3 See (2) in reply to report 2.

4 See (5) in reply to report 2.

5 See (6) in reply to report 2.

6 We added comments on both the PDF uncertainties which we assume to be at the level of current 13 TeV uncertainties and the EFT uncertainties which we do not include as we consider them to be uncertainties of the matching, and not of the dimension-six fit itself.

7 We compare the results of the 27 TeV fit to our more recent 13 TeV fit results in the conclusions (somewhat violating causality).

Thank you again for your very useful feedback.

Best regards,
Anke Biekoetter (on behalf of the authors)

---

## Round 1 · Referee Report · Anonymous (Referee 2) · 2019-1-4

Strengths

1- The main goal of the paper, i.e. deriving an estimate for the precision reach in a global Higgs couplings analysis of a HE-LHC machine is a timely and important one, in particular due to the recent activities of the HL/HE-LHC working group and as an input for the European Strategy Update in 2020. 2- The main results are clearly presented and summarised in a few simple ‘money plots’. 3- The analysis is robust, since it is mostly based on several previous works by the same group, which analysed present LHC data.

Weaknesses

1- While the choice of operators considered in the analysis, eq.(4), can be justified at present due to the fact that other operators entering the observables under study are strongly constrained by LEP data, this will not be anymore true by the time HE-LHC might run. The relevant operators are ‘contact term’-like which affect in a correlated manner $\bar{f} f V$, $\bar{f} f V V$, and $\bar{f} f V H$ couplings. In particular, the latter two couplings induce an $E^2$ enhancement in the high-energy tails of kinematical distributions. It has been shown in a series of recent works (e.g. [23] and [1807.01796]) that diboson and VH processes will be able to set stronger bounds than LEP on these operators by the end of HL-LHC. After HL-LHC these operators will have a constraint dominated by Higgs and diboson physics. Therefore, they should to be included in any analysis using these processes for prospects beyond HL-LHC, such as this one. The choice of the authors of not adding these operators can be maybe justified by the fact that in this preliminary analysis they relied on previous works to make it more robust, but this reduces the relevance of the results.

2- Another delicate aspect of the SMEFT analysis is the one of EFT validity. While the authors write that the analysis does not have any serious validity issue, the results seem to show otherwise. Indeed, from the plots of Fig.2 and 4 one can see that the EFT scale probed (for couplings f~1) is at most 2-3 TeV, while the energy reach in some of the high-energy distributions used in the analysis goes up to 5 TeV (Table 2). Clearly, then, the EFT interpretation is only valid for strongly-coupled theories where $f_x \gg 1$.

Report

In this paper the authors derive prospects of a HE-LHC machine running at 27 TeV for the sensitivity in Higgs couplings and some SMEFT Higgs-gauge operators. Such an analysis is very timely and an important input for future strategy decisions in high-energy physics.

The paper is very clear and the analysis robust. The two main issues are related to the limited choice of operators included in the SMEFT analysis, which I argue will not be the relevant one at the timescale of HE-LHC, and to the EFT validity, which implies that the analysis is valid only for strongly-coupled theories.
The second issue is a common one in EFT studies at the LHC and is not a problem specific with this paper. The one related to the choice of operators, instead, should definitely be addressed in a future work, however for a preliminary study such as this it can be accepted.

To summarise, I can recommend the paper for publication after the following changes are implemented.

Requested changes

1- The global Higgs analysis of Section 2 is analogous to those in terms of the $\kappa$ modifiers often presented by the experiments. In my opinion it makes sense to derive prospects for 27 TeV in this framework, particularly for the familiarity that experimental collaborations have with it. A more general bottom-up approach would instead require to include a larger set of deviations, also affecting differential distributions. I would then suggest to remove the sentence below eq.(2) ‘In principle, we could include a complete set of loop-induced operators,[…].’, since it uses the concept of loop-induced operators which is a debated one in the community and since in my opinion it is not necessary to justify the choice of terms in eq.(2).

2- The justification for not including Higgs self-coupling in eq.(2) given in page 4 is not correct in my opinion and should be changed. In fact, a general non-linear Lagrangian (beyond the one in eq.(2)) can describe changes in distributions: if the SMEFT can, then also the non-linear one can, with even more freedom. Still, with the motivation given above, I agree with the authors that it makes sense to add the Higgs self-coupling directly in the SMEFT analysis. This is also true since, as stated in Sec.4, HH production does not improve the limits on the parameters in eq.(2) and the Higgs self coupling does not affect in a relevant way single-production processes.

3- In page 4 it is written that for almost all couplings an improvement between 1.5/ab and 15/ab is observed. It should probably be stressed more that the improvement is much less than the factor ~3 one would expect if the limits would scale with the square-root of the luminosity, since these uncertainties are systematics and theory-dominated.

4- A table or equation with the definition of the operators in eq.(4) would help to make the paper more self-contained.

5- A discussion on the choice of operators, related to what written in 'Weakness-1', should be included in Section 3.

6- Following my comment on the EFT validity in 'Weakness-2', the related discussion in page 6 should be modified.

7- From some comments in the paper I understand that the analysis was performed including the quadratic terms in the EFT coefficients. This should be stated more clearly in Section 3.

  • validity: high
  • significance: high
  • originality: good
  • clarity: top
  • formatting: perfect
  • grammar: perfect

Author:  Anke Biekoetter  on 2019-01-17  [id 404]

(in reply to Report 2 on 2019-01-04)

Dear Referee,

Thank you very much for your report and your very useful feedback.

We have addressed your concerns in version 2 of our paper in the following way:

1 We have added equation (2) for the translation of the Delta framework into the kappa framework. Moreover, we removed the sentence on loop-induced operators.

2 We have rephrased the comment on the Higgs self-coupling saying that we will only consider it in the dimension-six fit as it will not affect single-Higgs production in a meaningful way.

3 We added the suggested comment.

4 We added the list of operators.

5 We added a discussion of the fermionic operators not included in our fit. We totally agree that they should eventually be included in a dimension-six fit.

6 We added a comment about the EFT only being valid if we do not see a peak in the diboson distributions.

7 We added the suggested comment.

Thank you again for your feedback.

Best regards,
Anke Biekoetter (on behalf of the authors)

---

## Round 2 · Referee Report · Anonymous (Referee 2) · 2019-1-22

Report

The authors addressed all the points raised in my first report.
They acknowledge that an analysis including operators with fermions would be advisable for the HE-LHC perspective, however I agree with them that it is beyond the scope of this paper.
For these reasons I recommend this paper for publication.

---

## Round 2 · Referee Report · Anonymous (Referee 1) · 2019-2-5

Report

The authors have addressed satisfactorily all the questions in my previous report, and I therefore recommend the article for publication.

---

## Editorial Decision

published